# African Mole-Rats May Have High Bone Conduction Sensitivity to Counterbalance Low Air Conduction Sensitivity

**DOI:** 10.3390/audiolres15030064

**Published:** 2025-06-03

**Authors:** Andrew Bell

**Affiliations:** Eccles Institute of Neuroscience, John Curtin School of Medical Research, Australian National University, Canberra, ACT 2601, Australia; andrew.bell@anu.edu.au

**Keywords:** mole-rats, family Bathyergidae, underground hearing, air conduction, bone conduction

## Abstract

**Background/Objectives**: Subterranean mole-rats live in an intricate system of underground tunnels, a unique acoustic environment that has led to adaptations to their hearing. Most experimenters have concluded that mole-rats have poor hearing thresholds, perhaps 20–40 dB less sensitive than rodents living on the surface. The potential problem identified here is that mole-rat thresholds have all been measured in air, whereas there is some evidence—theoretical and observational—to suggest that these animals may hear more sensitively via bone conduction. **Methods**: A wide-ranging review of the literature surrounding mole-rat hearing is undertaken and then interpreted in terms of the ways air conduction and bone conduction thresholds are measured. The important factor, often overlooked, is that the detection of an acoustic signal is most sensitive when there are matching impedances all along the transmission path, and the argument is made that, for subterranean mole-rats, more energy may be transmitted to their cochlea when the head is directly in contact with the earth than when an acoustic signal must propagate from the earth to the air and then reach the cochlea via the external and middle ear. **Results**: Based on observational evidence, theoretical considerations, and inferences from related species, the suggestion is made that, for African mole-rats, high bone conduction sensitivity could make up for their relatively poor air conduction thresholds. **Conclusions**: Bone conduction audiograms are needed for mole-rats, similar to those for other animals sensitive to substrate vibration such as snakes or amphibians. It is possible that the hearing thresholds of mole-rats may, when measured appropriately, be comparable to those of other rodents.

## 1. Introduction

African mole-rats (family Bathyergidae) have for decades been the subject of considerable scientific interest because of their hearing—or more particularly an apparent deficit in it. These burrowing animals, confined largely to the African continent, have consistently been found to have air conduction hearing thresholds less sensitive by as much as 25–50 dB compared to similar surface-dwelling rodents [1,2,3,4,5].

Their hearing has been called “vestigial” or “degenerate” [5,6,7], and a recent paper continues to reflect that view, describing the African mole-rat’s hearing as “poor” and of “low sensitivity” [8]. At the same time, the latter paper finds that mole-rats have levels of otoacoustic emissions that are comparable to, or even higher than, those reported in humans and chinchillas, a finding the authors consider remarkable—given that the mole-rat species under study had air conduction hearing thresholds at least 20 dB less sensitive. However, that same paper provides little discussion of how those conflicting attributes might be reconciled, does not mention bone conduction, and concludes with “it remains unexplained why these animals … that have a large number of hair cells within the fovea and produce levels of OAEs comparable to other species … have rather poor auditory thresholds”.

The purpose of the present communication is to draw attention to the paradox between elevated thresholds and robust OAEs, trace the origins of the problem, and suggest a resolution. It is proposed that mole-rats may possibly hear as sensitively as surface-dwelling rodents, and it is the methodology that has been used to test their hearing which is at fault. Most research to date has tended to overlook an important point: these underground-dwelling animals are adapted to detect sound (or substrate vibration more generally) by bone conduction, not air conduction, and yet published test protocols have invariably employed air-conducted stimuli [2,5,9,10,11]. It is hard to say why no-one has yet constructed a bone conduction audiogram for these animals, but perhaps it is the challenging nature of such experiments (see [12]).

If the data are reviewed from this perspective, it is possible that mole-rat hearing might be at least 20–40 dB more sensitive than previously stated (based on Figure 1 of [13], Figure 4.1 of [3], Figure 3 of [7], and Figure 5.4 of [14]), where the hearing thresholds of mole-rats and their surface-dwelling relatives are compared. That is, to take a specific case, perhaps the thresholds of naked mole-rats and gerbils, as shown in Figure 1, are comparable in terms of absolute hearing sensitivity. The suggestion is that the energy entering the cochlea may be about the same in both cases, although in one case (the gerbil) it is via air-borne sound while in the other (the naked mole-rat) it is through substrate-borne vibration.

Of course, mole-rats may well use a combination of air- and bone-conducted stimuli, as humans and many other vertebrates do, but it appears possible that experimentalists have, over the years, underestimated the hearing acuity of these animals. Nevertheless, a recent (2021) review of naked mole-rat biology [7] (pp. 125–126) largely agrees with the historical assessments and concludes that “the hearing of the naked mole-rat is indeed ‘degenerate’, in that several aspects of its audition have clearly deteriorated from what would reasonably be assumed to be the condition of its terrestrial ancestors”. The present paper disputes such a claim and provides a rationale for why fresh measurements are needed.

## 2. Air Conduction vs. Bone Conduction

A significant clue to resolving the mole-rat conundrum is to note that the ear-canals of these animals are long, narrow, and commonly filled with hair and cerumen [8,9,15], as illustrated in Figure 2. This state of affairs has been used to explain why African mole-rats respond so poorly to air-borne sound. But this restricted pathway needs to be seen in the context of other mammals whose ear canals are also blocked—whales, dolphins, and dugongs, for example—whose ear canals are packed with dense cerumen plugs [16]. Taking into account that cetaceans hear very well (up to ultrasonic frequencies) and that they too are mammals, an impediment to ear canal transmission can be interpreted as an important design feature, not a drawback. The successful transmission of sound relies on good impedance matching at all interfaces along the path [12] and, as in cetaceans, blocking the ear canal is a way by which bone-conducted sound can be augmented [17]. The same effect can be illustrated in humans: the sound of one’s own voice or the tapping of one’s teeth becomes much louder when the ear canals are blocked (for example, with the fingers).

By no means can cetaceans be called hard of hearing—they can hear over thousands of kilometres and detect ultrasonic echoes [16,18]—but instead of picking up sound conducted through their ear canals, whales and dolphins use a different strategy, which is bone conduction. In this mode of operation, underwater sound passes directly through the body—fat, tissue, and bone—to reach the middle ear cavity and then the inner ear. Relevant here is work on soft tissue conduction in humans [19,20] and on the hearing of amphibians, where sensitivity is needed in both air and water [21,22].

In a bone conduction scenario, the presence of air within the middle ear is crucial, for it presents an impedance discontinuity that allows interaction with the pressure component of a sound, which would otherwise pass straight through the animal. With an air cavity present, passing pressure waves cause a corresponding oscillation in volume, so that the cavity acts as a pressure-to-displacement converter [23]. The oscillating volume displaces the animal’s eardrum, which in turn moves the ossicular chain and stimulates the cochlea. The process resembles what happens with air conduction, except, of course, that in this case no ear canal is necessary. A good analogy is the air-filled swim bladder of certain fish which vibrates in response to underwater sound, vibration which is passed via the Weberian ossicles to the fish’s inner ear [23]. Another good demonstration of the principle is the helium-filled party balloon which, when placed in front of the mouth, vibrates strongly in response to speech (easily verified by touch) because of the impedance discontinuity created by the helium.

Note here that if an animal, such as a dolphin or whale, makes use of bone conduction and uses pressure-to-displacement conversion in the middle ear cavity, its ossicles may be light or heavy depending on which frequencies need to be detected. That is, the size of the ossicles is, by itself, not evidence that an air- or bone conduction route is favoured. Although some burrowing animals have heavy ossicles and seek to detect low-frequency sounds, mole-rats have high-frequency vocalisations and appear to have correspondingly low-mass ossicles. When Vice and colleagues considered the vocalisation question in the context of the naked mole-rat (*Heterocephalus glaber*) and its “poor hearing”, they pointed out that “It seems paradoxical to be highly vocal, but hard of hearing” [14] (p. 143). Elsewhere they note that “To put this into perspective, a human with this hearing profile [a 40 dB loss at 1–4 kHz] would be considered to have pathological hearing loss”. To make this specific, Barker and colleagues [24] list in their Table 6.1 the actual frequencies of naked mole-rat calls, which are 3–6 kHz for soft chirps, 2–5 kHz for whistles, and 3–7 kHz for trills. It indeed appears paradoxical for the animals to be effectively deaf to these frequencies—for which a 40 dB “loss” (compared to a gerbil) is displayed in Figure 1.

Humans hear by both air conduction and bone conduction [17,25], although normally most sound energy reaches the cochlea via air conduction. These two terms are not absolute, of course, in that even air-borne sound leads to some vibration of the bone of the skull. Nevertheless, the point made earlier is that blocking the ear canal is a prime way of augmenting bone conducted sound [26]. The inference to be drawn here is that mole-rats, and indeed possibly other burrowing animals, may be able to hear quite well via bone conduction, since earth-borne sound could potentially enter their bodies through points of contact with the surrounding burrows. Kimchi and colleagues performed experiments which suggested that blind mole-rats (*Spalax ehrenbergi*) detected seismic vibrations via both their jaws and their feet [27].

Evolutionarily, it is supposed that the ears of mole-rats have adapted to picking up vibrations traveling through the earth, with sound first reaching the body and then their ears. It has been documented that blind mole-rats “listen” by pressing their heads against the walls of their burrows [28], much like it is possible for a human to hear an approaching train miles away by touching the head to the rails. Likewise, blind mole-rats communicate their presence to others by thumping their heads against the roof of their tunnels [29,30]. It should be noted, however, that no head-pressing has been observed in naked mole-rats, although, as social animals, they do press against each other.

In summary, for a mole-rat, it may be easier to hear what is happening in its immediate underground environment (i.e., the acoustic signal is louder) if it presses its head against the wall of the tunnel, instead of listening to air-borne sound—especially if its ear canals are blocked. The enhanced bone conduction ability of mole-rats is speculation, but the point is that, despite previous suggestions along the same lines, no experiments to decide the question have yet been made.

Mason and coworkers have used the term “potentially degenerate” to describe the hearing of naked mole-rats [15,31], although they acknowledge that perhaps the term “adaptive” is more appropriate. They also point to two papers that strongly reject the “degenerate” label: one paper [6] argues, based on data with large scatter, that burrow acoustics resembles a sound-amplifying stethoscope, so that the animal might be in danger of having its air-conducted hearing overloaded (and so a reduction in sensitivity is adaptive rather than degenerate), while the other [13] argues the semantics of vestigial vs. adaptive. Regardless of semantics, however, all these papers share the view that mole-rats have insensitive hearing and only the work of Mason [15] mentions the possibility of bone conduction (based on earlier anatomical observations [26]).

Narins and colleagues have also explored the ways by which subterranean animals in general, including moles and mole-rats, pick up ground vibrations, emphasising the importance of bone conduction pathways and certain peculiar adaptations of ossicles [32,33,34]. Of interest, Narins draws attention to how previous work had not yet measured the sensitivity of mole-rats to substrate-borne vibration, and they predict that such work might constitute a “neuroethological gold mine” [35] (p. 645).

For humans, blocking the ear canals produces a broad hearing loss of air-conducted sound by roughly 40–50 dB [17], depending on frequency, but that does not make us deaf, as there is still the bone conduction route. Indeed, people with conductive deafness can benefit greatly by making use of it. A bone-conducted hearing aid (BCHA) picks up sound with an external microphone, conveys the signal to a bone-attached transducer, which then vibrates the skull [36]. In such a way, many hearing-impaired people have had their hearing sensitivity restored.

Given the above, it becomes possible to appreciate that using air conduction protocols to measure the hearing sensitivity of a mole-rat could fail to provide a complete picture of its hearing capacity—in the same way as it would for a dolphin or a snake [37]. Experimenters have generally put mole-rats in cages, placed loudspeakers some distance away, and measured the animals’ sensitivity to calibrated air-borne tones. Sometimes behavioral methods have been used to detect thresholds [5,38], at other times auditory brainstem responses (ABRs) [1,2,9,10], but in all cases the stimuli were delivered through the air.

No-one has yet placed a vibrator against the jaw of a mole-rat and measured its bone conduction thresholds. Such a technique has been applied to snakes [12], which also hear badly in air. If the same sort of tests were carried out on an African mole-rat, it is suggested that the results could be similar, with their bone conduction thresholds being much like the air conduction thresholds of their surface-dwelling relatives such as chinchillas, gerbils, and guinea pigs [3,7].

## 3. Other Methodological Problems

The foregoing has set out the main problem—that a distinction is needed between bone- and air-conducted sensitivity, especially in relation to high frequency sounds. But it is also worth noting other methodological errors that have tended to obscure the picture.

To begin, researchers have frequently used ABRs to determine thresholds [1,2,9,10]. The problem here is that the “threshold” determined in such a way is not a truly absolute measure: the technique depends on the experimenter (or an algorithm) discerning a wave peak above the noise, and this of course depends on the number of response waveforms averaged.

Whereas there is indeed a correlation between ABRs and behavioral thresholds, the latter is the only true threshold measure. However, to illustrate some of the substantial methodological problems involved, it is noted that Gerhardt and colleagues [10] find disparities of up to 35 dB between their ABR measurements and those of an earlier behavioural assessment, especially at high frequencies (Table 1 of [10]). Gerhardt and colleagues note a difference of 10–30 dB between the two techniques and they cast doubt on the behavioral method, citing unrepresentative responses and calibration difficulties.

An alternative method of assessing thresholds involves measurement of otoacoustic emissions. The difficulty here is that, for the strongest signal, an acoustic probe needs to be placed as close to the eardrum as possible, 6 mm into the 8 mm ear canal in the case of Kössl and colleagues [39], but only 2–4 mm deep in the case of Pyott and coworkers [1]. Calibration in such a situation is difficult, so Kössl and colleagues chose to measure the level that produced an arbitrary −10 dB of distortion in distortion product otoacoustic emissions (DPOAEs). However, this meant there was no absolute measure of sensitivity in terms of sound pressure level (SPL), although they were still able to conclude that mole-rat hearing was “comparable” to those in other laboratory animals. However, this result has not swayed the general opinion that mole-rats have compromised hearing.

Pyott et al. [1] chose not to clear out the cerumen and inserted the DPOAE probe only partly into the ear canal. They failed to pick up a microphone signal above the noise, even at 70 dB SPL, and on this basis they concluded that the animal had, “counterintuitively”, poor hearing and that, “remarkably”, cochlear amplification was deemed absent. More prosaic explanations could be that the ear canal was blocked with cerumen, noise levels were too high, or that anesthetics had depressed OAEs (possibilities set out by Manley and colleagues [8], a paper which also reports some cases of poor or absent responses likely due to the “access problem” (p. 5)).

Nevertheless, when Manley and colleagues [8] placed their OAE probe 1 mm into the ear canals of three different species of African mole-rats, they were able to record “SFOAE levels … comparable to those in humans and chinchillas” (p. 5). They called this finding “remarkable” because the animals’ hearing thresholds were, in terms of accepted understanding, “at least 20 dB poorer”. Of course, this last statement needs to be qualified by “in air”. The authors did not consider the possibility that mole-rats may be able to detect an acoustic signal much more sensitively via bone conduction.

## 4. Resolving a Long-Standing Issue

A final curious aspect of this story is that the possibility of mole-rats making use of bone conduction was explicitly put forward more than 30 years ago based on the anatomical arrangement of the middle ear, lower jaw, and skull [28]. Rado and colleagues pointed to the “jaw listening behavior” of blind mole-rats in which the animals press their cheeks and lower jaws against the wall of a tube enclosing them, apparently “listening”. They also commented on the similarity to dolphins and reptiles who also receive sound through their jaws. A few years later, Rado and coworkers gave detailed demonstrations that the blind mole-rat senses vibrations via its auditory system [40,41], countering an earlier suggestion [42] that detection might involve the somatosensory system (i.e., tactile sensing). A subsequent paper by Kimchi and colleagues [27] endeavoured to show that the animal detects vibration through its paws, but of course this does not rule out that the paws are part of a soft tissue and bone conduction pathway [19,20]. In any case, the authors concluded that both bone conduction hearing and somatosensory perception might both be at work.

Historically, there seems to have been a reluctance to take the bone conduction idea seriously. For example, even though Heffner and Heffner [38] cite the earlier paper of Rado and colleagues and its bone conduction proposal [28], the later work still continued to measure the hearing of the mole-rat using air-conducted sound delivered via a loudspeaker. As a partial acknowledgement, they did take the “precaution” of placing their loudspeaker on a layer of foam to prevent vibrations reaching the animals through the floor. In like manner, other researchers have taken similar precautions, not just isolating their loudspeakers off the floor [5,10] but also mounting the holding cage on foam rubber [5]. Nevertheless, actual measurements of vibration sensitivity have not yet been made.

The author is unable to explain why, more than 3 decades later, this situation persists, but it is hoped that the current paper may help renew interest in the bone conduction idea.

As mentioned earlier, Narins, Mason, and colleagues had already drawn attention, in work between 2001 and 2010, to the possibility that bone conduction operates more generally in the ears of burrowing animals [26,32,34]. They focused attention on the golden mole, which, although in a taxonomically different family (Chrysochloridae), is an animal that, like the mole-rat, spends a lot of its life underground. Based on the mole’s hypertrophied ossicles, these investigators proposed that the animal and its subterranean cousins detect substrate vibrations by inertial bone conduction [26,32,33,34,43]. The most detailed consideration of this mode of hearing, as applied to three families of mole-rats, is set out in [26], where three possible mechanisms are explicitly set out: (1) inertial bone conduction via the ossicles; (2) radiation of bone-conducted sound into the external ear canal; and (3) a fluid pathway not involving the middle ear. It is worth noting that this list omits the well-recognised “pressure-to-displacement” conversion as used by cetaceans—which involves radiation of sound from inside the middle ear cavity [23]. As already noted, such an arrangement involves the middle ear cavity acting as an impedance discontinuity within the body of the animal and, like a swim bladder or party balloon, giving rise to vibration of the eardrum (followed by transmission via the ossicles to the cochlea). In this picture, it is wrong to think that bone conduction requires the mass of the ossicles to be large (c.f. [15], p. 20), and in fact a smaller mass allows higher-frequency hearing. The conclusion reached here is that an appreciation of a pressure-to-displacement mechanism within the middle ear cavity itself (even if the ear canal is blocked) makes it possible to bring together two aspects of their hearing: a bone conduction sensitivity extending to high frequencies.

Of course, it should be emphasised that bone conduction is a complex process involving multiple possible pathways, and is incompletely understood [17,26]. Thus, in any particular species, whether mole-rat or cetacean, it is unclear which route predominates. In the case of mole-rats, more investigation is needed to decide whether they make use of pressure-to-displacement conversion within the middle ear cavity or employ some other bone conduction mechanism. However, irrespective of the pathway, the major message of this paper is to draw attention to the possibility that these animals could use enhanced bone conduction to counterbalance their impaired air conduction hearing.

The implication of bone conduction for naked mole-rats—highly vocal, social animals which huddle together—is that they are able to both utter and hear their own soft chirps at 3–6 kHz [24]. A list of all the different call types made by these animals, along with their frequencies, is given in Table 6.1 of [24], and it is suggested that all these vocalisations—up to 8.1 kHz—are audible to conspecifics (unfortunately the table gives no measure of corresponding sound intensities, but the descriptor ‘soft’ offers an important clue).

A good analogy is to the hearing of cetaceans, animals whose ancestors once lived on land but returned to the ocean, adapting their hearing apparatus so that they could hear sensitively in a radically different acoustic environment—underwater. To hear effectively, they needed to make use of bone conduction, and a likely mechanism is a pressure-to-displacement mechanism inside the bulla. Likewise, the ancestors of mole-rats once lived on land, but they moved to a different habitat—underground. Again their hearing had to adapt to deal with this novel acoustic environment, and the possibility raised here is that sound detection may also involve a similar bone conduction mechanism based on pressure-to-displacement conversion in the middle ear.

## 5. Conclusions

Despite sporadic investigation into the anatomy of mole-rat ears—work which has helped keep the bone conduction idea in view [15,44]—this mode of hearing has continued to be neglected in psychophysical studies. Yet some anatomical and behavioral observations—largely circumstantial—suggest that African mole-rats, and possibly other burrowing mammals, achieve the lowest auditory thresholds via bone conduction. The possibility was raised more than 30 years ago, but unfortunately it still remains untested. Mole-rat hearing may not be poor, and in fact they may combine both modalities and hear appreciably better than we have assumed. Detailed experiments are required to clearly establish the facts of the matter.

## Figures and Tables

**Figure 1 audiolres-15-00064-f001:**
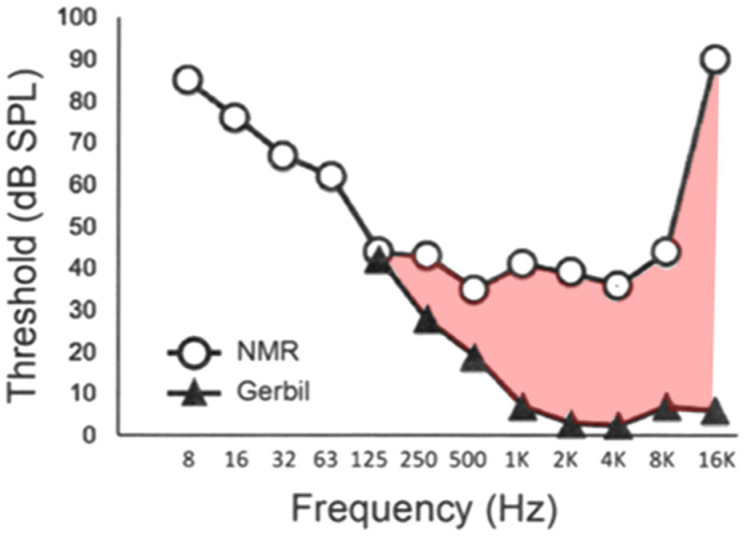
A comparison between the hearing thresholds (by air conduction) of the naked mole-rat (NMR) and of the gerbil. Note that when measured in air, the NMR appears to have a 40 dB loss in hearing sensitivity (pink) at high frequencies compared to its surface-dwelling relative. The hypothesis put forward here is that both animals have roughly the same absolute sensitivity (expressed on a common energy flow scale), but that the mole-rat may employ some bone conduction mechanism—currently unidentified, but perhaps similar to that of cetaceans—to achieve comparable sensitivity at higher frequencies. Adapted from Figure 3 of [7] (used under a Creative Commons CC-BY-NC-ND license).

**Figure 2 audiolres-15-00064-f002:**
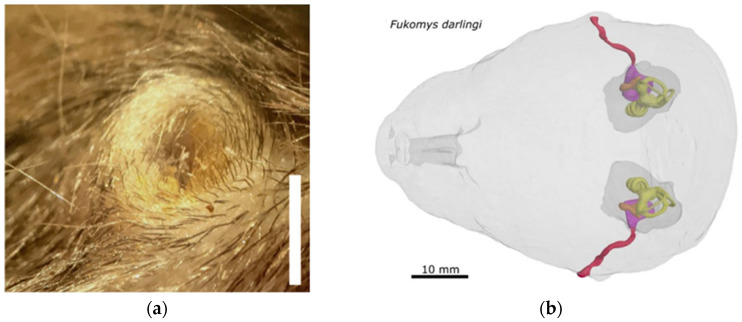
(**a**) The outer ear of the Mashona mole-rat, *Fukomys darlingi*. There is no external pinna, and the canal is filled with hair and cerumen. Scale bar = 2 mm. (**b**) Surface rendering of a micro-CT image of the ears and skull of this species, showing the long, narrow, and winding ear canals (red, cartilage-lined; magenta, bony) leading to the eardrums and ossicles (orange) and inner ears (yellow). Note the air-filled middle ear cavity (bulla, dark grey). The cochleas are embedded within the bone of the skull. (**a**) from [9] and (**b**) from [8]; both reproduced with permission via the Copyright Clearance Center.

## Data Availability

No new data were created or analyzed in this study. Data sharing is not applicable to this article.

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
