# Peer review of "African Mole-Rats May Have High Bone Conduction Sensitivity to Counterbalance Low Air Conduction Sensitivity"

_audiolres, 2025, doi:10.3390/audiolres15030064_

Round 1
Reviewer 1 Report
Comments and Suggestions for Authors
In general, the manuscript is well written, and the scientific dispute that the author raises is reasonable. But I have a few comments that I hope the author can address in the revision.
Line 31-40. A possible explanation for mole rat’s poor air-conduction hearing is that its outer and middle ear may transmit sound not as efficiently as in other species. I don’t see how this theory can be completely ruled out.
Line 83-85. It’ll be better to elaborate the plug-in effect on BC hearing here to explain why a blocked ear canal could benefit hearing.
107-108. I am not sure if the pressure-to-displacement mechanism would work for mole rats. Such mechanism may work well if the middle-ear cavity is relatively large compared to wavelength and the wall of the middle ear is not so rigid. Assume the sound speed in the soil is 1000 m/s (it’s a reasonable number since sound speed in rigid metal is only 3000-5000 m/s), then the wavelength of a 2000 Hz sound is half meter long, which is way too big compared to the size of the animal’s middle ear. And the middle-ear wall of the animal is unlikely to be soft. It’s unlikely that the mole rat’s middle ear would be compressed at a meaningful magnitude in the presence of underground vibrations. I think the ossicles’ inertia is a better candidate for driving the cochlea.
114-117. In theory, the ossicles have to favor their resonant frequencies, and the resonant frequencies for air conduction are almost identical to those for bone conduction. This is backed by experimental work in human and chinchillas.
Line 170-171. Given the size of the animal, it’s hard to believe their vocal and hearing have a low-frequency emphasis.
Section 3. How about the ABR and DPOAE test results from other species? Those measurements may be challenging in other small rodents too. If such results exist, do they suggest the hearing in other animals are better than the mole rat?
Line 263-266. As commented above, I disagree with the pressure-to-displacement mechanism.
Reviewer 2 Report
Comments and Suggestions for Authors
The paper is interesting, but there are some points that needs a revision.
- I think the title must be changed. I find the concept of 'hearing better' somewhat misleading in this context. A higher bone conduction sensitivity does not necessarily imply better hearing, but rather a different auditory processing mechanism or pathway.
- the abstract needs to be written more accurately
- when you say “hearing thresholds” in the text, like in line 29, you must specify which testing or way (air-bone) of testing are you referring.
- the concept of “hear as well” in line 43 is not described well and has a wrong scientific base. If you have a good bone conduction and poor air conduction, it doesn’t mean that you hear well like some other animals.
- Why do you say that NMR and gerbil have comparable in terms of absolute hearing sensitivity? They have completely different hearing thresholds.
- In line 121 you need to better describe the idea, because yes you have an increment of loudness, but a loss in discrimination, so this not results in a better hearing.
- the concept in lines 171 to 179 is interesting, but something need to be added taking in mind the idea of pathological hearing loss: we define hearing loss comparing the threshold in the same species, so I define a man like having hearing loss because a threshold above 25 dB nHL, so if all the NML have around the same threshold, you cannot define an individual like having a pathological hearing, but you can consider it like different. I think that your idea is that they have a “different” way of hearing, but this need to be more clear in the text throughout all the review. Furthermore, could be interesting to analyse the sound they use to communicate, because they could have a particular vibration pattern that is received and perceived better, so the message is not the air component, but the vibration-bone one.
- About lines 214 to 219 in literature is described the registration of SFOAE till threshold of 40-50 dB HL, so this could be the reason why them could be sometimes found in NMR.
- In the Conclusions section you can better state the idea that by testing the bone conduction we could have a better understanding of his hearing, because is not foregone that you’ll found a better bone threshold and this must be understated.
- There’s not [35] in the text
- It would be better to have a more recent references, if available
Round 2
Reviewer 1 Report
Comments and Suggestions for Authors
The author has addressed the previous comments adequately. I only have one comment.
Line 284-305. I agree with the author that the contraction of the middle ear cavity could exist and contribute to BC hearing. But how much is its contribution in comparison to the other, classical BC mechanisms? I am afraid that no one has the answer at the moment. This paragraph seems to overly emphasize the pressure-to-displacement mechanism without carefully considering the other mechanisms, and I think that is inappropriate. I suggest the author make a more balanced discussion here.
